# Active Learning for Flow Matching Model in Shape Design: A Perspective from Continuous Condition Dataset

## Abstract

Although the flow matching model has demonstrated powerful capabilities in modern machine learning, its training notoriously relies on an incredibly large scale of high-quality labeled samples. Nevertheless, the acquisition of high-quality labeled datasets is hindered by exorbitant labeling costs in certain fields, notably medical imaging and numerical simulation. Therefore, selecting the most informative samples for training at minimal cost poses a key challenge in these fields. This issue constitutes a central topic in active learning, a subfield of machine learning dedicated to maximizing model performance while minimizing annotation cost. The central challenge involves developing an optimal query strategy to acquire the most informative data samples with minimal labeling effort. This paper presents a pilot study that investigates the application of active learning, which traditionally explored within the context of discriminative models, to flow matching models. By analyzing flow matching models through a piecewise-linear neural network framework, this work elucidates how individual data points influence the diversity and accuracy of the model. Leveraging this analytical framework, we propose two distinct query strategies: one aimed at enhancing model diversity, and the other designed to improve model accuracy. We demonstrate that these two strategies are inherently conflicting, providing a partial explanation for the fundamental trade-off between diversity and accuracy in flow matching models from a dataset perspective. Furthermore, we introduce a mixed strategy that combines both strategies through a weighted mechanism, enabling adjustable control over the diversity-accuracy trade-off by tuning the corresponding weights. Extensive experiments validate the effectiveness of our approach, showing that the proposed query strategies outperform those designed for discriminative models.

## 1 Introduction

Recently, flow matching models achieve state-of-the-art performance in image and various other generating tasks (Dhariwal & Nichol (2021); Ho et al. (2022); Saharia et al. (2022)) and are one of the fundamental building blocks of the more advanced image and video synthesis systems, e.g., DALL-E-3 (Ramesh et al. (2022)) and Veo3 (Esser et al. (2023)). The success of these models is attributed primarily to the availability of large-scale, high-quality labeled training datasets.

However, the acquisition of high-quality labeled datasets is notoriously challenging in some domains due to exorbitant annotation costs. This is particularly true in fields like medical imaging Budd et al. (2021) and numerical simulation Wu et al. (2024), where the cost of obtaining labels far exceeds that of data acquisition. For instance, in medical imaging, the cost of annotating images by expert radiologists significantly exceeds the initial image acquisition cost. Similarly, in automotive engineering, while generating raw simulation models is relatively inexpensive, obtaining high-fidelity numerical simulation results, which require extensive validation and expert interpretation, entails substantially greater effort and expense. So a fundamental challenge in these fields is to select the most informative samples for labeling while minimizing cost. This problem defines the core mission of active learning, a machine learning subfield dedicated to maximizing model performance under constrained annotation resources by developing optimal query strategies.

The most common Active Learning strategies include uncertainty based sampling Ren et al. (2021); Li et al. (2024), query by committee Seung et al. (1992), and representation-based sampling Geifman & El-Yaniv (2017); Sener & Savarese (2017), etc. The core principle guiding these methods is to identify and query the most valuable samples to improve the model's decision boundary. Meanwhile, a parallel research direction explores the integration of generative models within the active learning framework. For example, GAAL Zhu & Bento (2017); Lan et al. (2024) proposed employing generative networks for data augmentation. However, its randomly generated samples do not necessarily yield higher informativeness than those in the original dataset. In contrast, BGADL Tran et al. (2019) simultaneously trains both a generative network and a classifier to produce samples within uncertain or disagreement regions. Subsequent methods, including VAAL Sinha et al. (2019) and TAVAAL Kim et al. (2021), further extended this concept by leveraging adversarial learning frameworks to enhance data augmentation and improve feature representation. However, these methods primarily focus on "generative models for active learning", rather than "active learning for generative models". In other words, their main objective is to boost the performance of discriminative models. Consequently, active learning specifically designed for generative models has received limited attention. For example, GALISPZhang et al. (2024) consider "subject of interest" which transforming the open querying problem in the label space into a semi-open one. Specifically, they design and test algorithms on a set of specific labels rather than in the entire label space.

In this paper, we discuss the generalization error of generative models in a manner analogous to that of discriminative models Sugiyama (2015). Specifically, we focus on the generation results across the entire condition space rather than under specific conditions. To conduct such analysis, we propose an analysis framework based on piecewise-linear neural networks Montúfar et al. (2014); Goujon et al. (2024), which helps us analyze the generation results of flow matching models. Specifically, we assume the flow matching model's neural network is piecewise-linear.

Analyzing the generalization performance of closed-form flow matching models Scarvelis et al. (2023); Chen (2025) by this framework, we establish the generalization mechanisms of flow matching models and obtained the pattern of how data affects diversity and accuracy. Our analysis reveals that data with the same label in the dataset contributes to the diversity of the model, while data with different labels in the dataset contributes to the accuracy of the model. Our findings elucidate the fundamental diversity-accuracy trade-off inherent in dataset composition. Guided by this insight, we formulate two targeted sampling strategies designed to augment diversity and accuracy individually. Furthermore, we demonstrate that a weighted integration of these antagonistic strategies provides a practical means to navigate this trade-off and balance both performance metrics.

Finally, we evaluated our query strategies on a synthetic dataset and three real-world shape design tasks. Shape design is an application of generative models. In this context, models are given continuous performance requirements (acting as labels) and are tasked with producing a corresponding design shape Heyrani Nobari et al. (2021). In addition, numerical solvers are used to accurately obtain labels for generated shapes, eliminating the need for manual annotation. The results demonstrate that our query strategy surpasses classical strategies designed for discriminative models in achieving either diversity or accuracy. Moreover, by strategically weighting these query strategies, we enable the formulation of tailored approachs that navigate the trade-off between diversity and accuracy.

The key contributions of this work are summarized as follows:

1) **Flow Matching Model Analysis Framework**: We introduce a novel analytical framework for flow matching models that leverages piecewise-linear neural networks and closed-form flow matching models, enabling rigorous theoretical characterization. This approach elucidates how individual data points influence the model's diversity and accuracy.

2) **Efficient Query Strategy for Active Learning**: Leveraging the proposed analytical framework, we present a pilot study on the application of active learning to flow matching models, introducing two novel query strategies: one aimed at enhancing model diversity and the other at improving model accuracy. These strategies represent competing objectives, underscoring the inherent trade-off between diversity and accuracy from a data-centric perspective.

3) **Experimental Validation**: Experiments on multiple datasets demonstrate that the two proposed query strategies outperform the direct use of standard active learning method designed for discriminative models in terms of diversity and accuracy, respectively. Additionally, a weighted combination

of the two strategies can be formed to create a hybrid query approach, allowing for a tunable trade-off between diversity and accuracy by adjusting the corresponding weights.

## 2 METHODOLOGY

### 2.1 PROBLEM DEFINITION

In the pool-based active learning method, we define $U^n = \{\boldsymbol{X}, \boldsymbol{Y}\}$ as an unlabeled dataset with $n$ samples where where $\mathbf{x} \in \boldsymbol{X}$, $\mathbf{y} \in \boldsymbol{Y}$. $L^m = \{\boldsymbol{\mathcal{X}}, \boldsymbol{\mathcal{Y}}\}$ is the current labeled training set with $m$ samples, where $\boldsymbol{x} \in \boldsymbol{\mathcal{X}}$, $\boldsymbol{y} \in \boldsymbol{\mathcal{Y}}$. Our goal is to design a query strategy $Q_D$ ($U^n \xrightarrow{Q_D} L^m$) to maximize the diversity score of the model, and a design query strategy $Q_A$ ($U^n \xrightarrow{Q_A} L^m$) to maximize the accuracy score of the model.

### 2.2 PIECEWISE-LINEAR ANALYSIS FRAMEWORK

In this paper, we leverage specific characteristics of neural networks to analyze the flow matching model, rather than analyzing the complex networks themselves. Particularly, this investigation centers on continuous and piecewise-linear neural networks (CPWL NNs)Montúfar et al. (2014); Goujon et al. (2024). The fundamental concept is that the neural networks can be formulated as piecewise-linear functions. Furthermore, researchers investigated the condensation phenomenon of neural networkLuo et al. (2021); Xu et al. (2025). They pointed out that under certain conditions, such as when using dropout or small initialization, the parameters of neural networks may undergo condensation. This means that after fully fitting the dataset, the network tends to reduce the number of effective parameters while also decreasing the number of inflection points. As a result, the network exhibits piecewise-linear interpolation behavior. In this paper, we hypothesize that neural networks employed in flow matching also exhibit the property of piecewise-linear interpolation. Specifically, when condition $\boldsymbol{c}_0$ in the labels of the dataset, the flow field of the closed-form flow matching modelScarvelis et al. (2023); Chen (2025), when consider the optimal transmission noise schedule Lipman et al. (2022):

$$\boldsymbol{u}_t(\boldsymbol{x}', \boldsymbol{c}_0) = \frac{\sum_i^m p_{t,i} \boldsymbol{e}_{t,i}}{\sum_i^m p_{t,i}} \tag{1}$$

where $\boldsymbol{e}_{t,i}$ is the noise that make $\boldsymbol{x}_i$ to $\boldsymbol{x}'$, $\boldsymbol{x}_i$ is the data with label $\boldsymbol{c}_0$ in the dataset, $m$ is the number of the data with label $\boldsymbol{c}_0$ in the dataset, $p_{t,i}$ is the probability density of $\boldsymbol{e}_{t,i}$. Eq1 means the vector field is a linear combination of data in a dataset.

When condition $\boldsymbol{c}^*$ is not in the labels of the dataset, the output of the neural network is defined as the interpolation of the the output of the neural network of the conditions near $\boldsymbol{c}^*$:

$$\boldsymbol{u}_t(\boldsymbol{x}', a_0\boldsymbol{c}_0 + a_1\boldsymbol{c}_1 + ... + a_k\boldsymbol{c}_k) = a_0\boldsymbol{u}_t(\boldsymbol{x}', \boldsymbol{c}_0) + a_1\boldsymbol{u}_t(\boldsymbol{x}', \boldsymbol{c}_1) + ... + a_d\boldsymbol{u}_t(\boldsymbol{x}', \boldsymbol{c}_d) \tag{2}$$

where $\boldsymbol{c}^* = a_0\boldsymbol{c}_0 + a_1\boldsymbol{c}_1 + ... + a_d\boldsymbol{c}_d$. $a_0$, $a_1$, and $a_d$ are interpolation coefficients. $\boldsymbol{c}_0$, $\boldsymbol{c}_1$ and $\boldsymbol{c}_d$ are the labels that exist in the dataset. $\boldsymbol{c} \in \mathbb{R}^z$, $z = d$, the label space is divided into several sub regions, each sub region being a convex hull with $d+1$ vertices. $[a_0, a_1, ..., a_d]$ can be easily calculated using the label $[\boldsymbol{y}_i, \boldsymbol{y}_j, ..., \boldsymbol{y}_k]$ of $[\boldsymbol{x}_i, \boldsymbol{x}_j, ..., \boldsymbol{x}_k]$. Because $d + 1$ points form a $d$-dimensional plane.

Under this assumption, for any given condition $\boldsymbol{c}$ ($\boldsymbol{c}$ exists in the dataset), the flow matching model is constrained to output only the corresponding sample from the dataset Gu et al. (2023). Besides, by using $Lemma1$ proven in Appendix A, we know that the vector field in Eq2 will result in the generated sample $\boldsymbol{x}^*$, being an interpolation of $\boldsymbol{x}_i$, $\boldsymbol{x}_j$, $\boldsymbol{x}_k$, etc.

$$\{\boldsymbol{x}^* | \boldsymbol{x}^* = a_0\boldsymbol{x}_i + a_1\boldsymbol{x}_j + ... + a_d\boldsymbol{x}_k\} \tag{3}$$

where $\boldsymbol{x}_i$ is data with label $\boldsymbol{c}_0$ in the dataset, $\boldsymbol{x}_j$ is data with label $\boldsymbol{c}_1$ in the dataset, $\boldsymbol{x}_k$ is data with label $\boldsymbol{c}_d$ in the dataset, etc.

Eq3 provides the generation law of the closed-form piecewise-linear flow matching model. Specifically, interpolation in the label space results in corresponding interpolation in the data space as illustrated in Fig1a. It is worth noting that the label dimension $d$ is generally smaller than the data dimension, meaning that the interpolation coefficients derived from the labels induce lower-dimensional interpolation in the data space. As a methodological note, the generated samples are provided without accounting for their respective generation probabilities. The probability of certain samples can be very small and even zero, as it is inherently contingent on the input condition, labels, and the characteristics of the data distribution. Therefore, Eq3 establishes an upper bound on the diversity of generated samples.

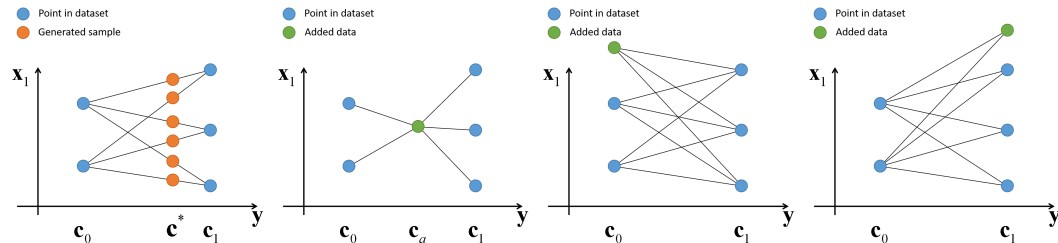

(a) The generated samples at $\boldsymbol{c}^*$

(b) Adding new point between $\boldsymbol{c}_0$ and $\boldsymbol{c}_1$.

(c) Adding new point at $\boldsymbol{c}_0$.

(d) Adding new point at $\boldsymbol{c}_1$.

Figure 1: Comparison of different point addition strategies and their generated samples, with the black line indicating the possible generated samples.

## 2.3 DIVERSITY FROM DATASET

Building upon the analysis in the previous subsection, we have derived the generation rules of the flow matching model. This understanding facilitates the analysis of how specific data points affect the resulting generated samples. Inspired by the method for estimating the number of samples a model can retrieve from a dataset Dombrowski et al. (2025), we quantify the number of individual sample points that the model can generate under a given condition $\boldsymbol{c}^*$. Specifically, we propose to increase the number of such individual samples as a query strategy.

For the sake of simplicity, consider the case of $\boldsymbol{c} \in \mathbb{R}^1$ and $d = 1$. As shown in Fig1a, when $\boldsymbol{c}^* \in (\boldsymbol{c}_0, \boldsymbol{c}_1)$, and there are $m$ samples labeled as $\boldsymbol{c}_0$ and $n$ samples labeled as $\boldsymbol{c}_1$ in the dataset (no data labeled between $\boldsymbol{c}_0$ and $\boldsymbol{c}_1$), the maximum generated sample type under condition $\boldsymbol{c}^*$ is $mn$. As shown in Fig1b, when adding a new data with label $\boldsymbol{c}_a$ ( $\boldsymbol{c}_0 < \boldsymbol{c}_a < \boldsymbol{c}_1$) to the dataset, the interval $(\boldsymbol{c}_0, \boldsymbol{c}_1)$ is divided into two segments $(\boldsymbol{c}_0, \boldsymbol{c}_a)$ and $(\boldsymbol{c}_a, \boldsymbol{c}_1)$. When $\boldsymbol{c}^* \in (\boldsymbol{c}_0, \boldsymbol{c}_a)$, the model will generate up to $m$ types of samples, and when $\boldsymbol{c}^* \in (\boldsymbol{c}_a, \boldsymbol{c}_1)$, the model will generate up to $n$ types of samples. Compared to the original dataset, this point adding strategy reduces the number of types of points at each $\boldsymbol{c}^*$, thereby decreasing the diversity of the model. Therefore, to increase the diversity of the model, we can only consider adding data labeled $\boldsymbol{c}_0$ or $\boldsymbol{c}_1$. As shown in Fig1c, adding data points labeled $\boldsymbol{c}_0$ will result in the model generating up to $(m + 1)n$ types of samples under condition $\boldsymbol{c}^* \in (\boldsymbol{c}_0, \boldsymbol{c}_1)$. While as shown in Fig1d, adding data points labeled $\boldsymbol{c}_1$ will result in the model generating up to $m(n + 1)$ types of samples under condition $\boldsymbol{c}^* \in (\boldsymbol{c}_0, \boldsymbol{c}_1)$. Obviously, to increase the number of types of points, we need to balance the number of data labeled $\boldsymbol{c}_0$ and $\boldsymbol{c}_1$ in the dataset.

Through the aforementioned analysis, we can design a query strategy $Q_D$ that increases model diversity:

$$Q_D = \underset{\mathbf{x} \in X}{\arg\max} - \alpha \, distance(\mathbf{y}, \boldsymbol{\mathcal{Y}}) + \beta \Delta entropy + \gamma \, distance(\mathbf{x}, \boldsymbol{\mathcal{X}}) \tag{4}$$

where $\alpha, \beta, \gamma$, are weighting coefficients. $\Delta entropy$ means the entropy increase of labels brought by new labels. $distance$ means distance from data point to dataset, we chose the minimum Euclidean distance in the experiments. Specifically, the minimum distance between a data point and all points in the dataset.

The $Q_D$ comprises 3 terms, the first term $-distance(\mathbf{y}, \boldsymbol{\mathcal{Y}})$ encourages new data points to have labels similar to those in the existing dataset. The preceding analysis prescribes that the labels of new data must be strictly identical to those already in the dataset. However, obtaining such exact matches is typically infeasible in practice. Accordingly, we impose the weaker condition that the labels of new data exhibit sufficient similarity to the labels present in the dataset. For unlabeled data, we employ Radial Basis Function (RBF) Neural Networks for label prediction due to their favorable optimization properties. The second term $\Delta entropy$ encourages the new data points to promote a more uniform label distribution across the dataset. This entropy corresponds to classification entropy, rather than being computed directly as information entropy. Specifically, we first partition the dataset labels into clusters and then compute the entropy of the label distribution across these clusters. A cluster is defined as a set of data points whose inter-point distances fall below a given threshold. The last term $distance(\mathbf{x}, \boldsymbol{\mathcal{X}})$ is inspired by the coreset concept Sener & Savarese (2017). It encourages the query strategy to select new data points that are farther from the existing dataset once the first two conditions are satisfied, thereby avoiding duplicating data and improving diversity.

## 2.4 ACCURACY FROM DATASET

Through the analysis in subsection 2.2, it can be concluded that interpolation in the condition space leads to corresponding interpolation in the data space. Furthermore, $Lemma2$ provides the error bound of the model within a subregion, given by:

$$|f(\boldsymbol{x}^*) - \boldsymbol{c}^*| \leq Kmax||\boldsymbol{c}_i - \boldsymbol{c}_j||^2 \tag{5}$$

where $f(\boldsymbol{x}) = \boldsymbol{y}$ represents authentic labels, $K$ is related to $f$ and $d$. $\boldsymbol{c}^*$ is the condition, $\boldsymbol{x}^*$ is the generated sample generated by the model given $\boldsymbol{c}^*$. $max||\boldsymbol{c}_i - \boldsymbol{c}_j||^2$ means the maximum distance of any two points in the subregion of label space.

In Eq5, the upper bound on the error within each subregion is determined by the maximum distance between any two points in the subregion. To reduce the error upper bound, a natural approach is to minimize this maximum distance. Accordingly, within the query strategy aimed at enhancing model accuracy, it is intuitive to select new data points whose labels are farthest from those already present in the dataset, as illustrated in Eq6. Essentially, $Q_A$ performs the coreset algorithm Sener & Savarese (2017) in the label space.

$$Q_A = \arg\max_{\mathbf{x} \in \boldsymbol{X}} distance(\mathbf{y}, \boldsymbol{\mathcal{Y}}) \tag{6}$$

For unlabeled data, we employ Radial Basis Function (RBF) Neural Networks to infer their corresponding labels. Upon comparing Eq4 and Eq6, it becomes apparent that the two strategies exhibit a fundamental conflict: $Q_D$ aims to seek new samples with $distance(\mathbf{y}, \boldsymbol{\mathcal{Y}})$ being smaller, while $Q_A$ aims to seek new samples with $distance(\mathbf{y}, \boldsymbol{\mathcal{Y}})$ being larger. In other words, data sharing the same label enhance the model's diversity, whereas data with distinct labels improve its accuracy. This clarifies why diversity and accuracy represent a trade-off from the perspective of dataset composition. Furthermore, Eq4 and Eq6 do not incorporate the trained flow matching model, but instead operate directly on the dataset for data selection. This implies that the available annotation budget can be utilized efficiently by training only the RBF neural networks for label prediction, thereby avoiding the need for repeated training of the flow matching model.

Considering that Eq4 solely enhances model diversity while Eq6 only improves model accuracy, a natural extension is to combine these two query strategies to balance the trade-off between diversity and accuracy. This leads to:

$$Q_{hybrid} = \omega Q_D + (1 - \omega)Q_A \tag{7}$$

where $\omega$ controls the ratio of $Q_D$ to $Q_A$.

As shown in Fig2, the dataset is unevenly distributed in both the data space and the label space. Different query strategies lead to the selection of different new data points. In particular, the coreset method selects data points that ensure a more uniform coverage of the data space. The committee

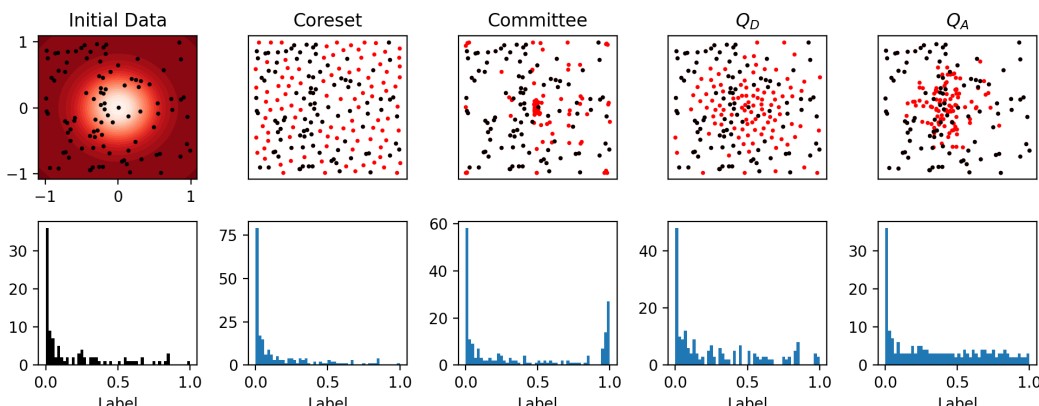

Figure 2: Comparison of different query strategies. Three-quarters of the data points are evenly located on the left side, while the remaining quarter are on the right. $f(\boldsymbol{x})$ is the Gaussian distribution with $\boldsymbol{\mu}$ at $[0, 0]$. The first row depicts the data distributions, and the second row shows the corresponding label distributions. In all subfigures, the black points represent the initial data, while the red points represent the new data selected by each query strategy.

method selects new data with the greatest output discrepancy among prediction models. Consequently, the selected samples tend to cluster at the edges of the label distribution, such as the regions corresponding to labels $0$ and $1$. This divergence arises from the distinct extrapolation strategies employed by the different prediction models. $Q_D$ selects new data by ensuring a uniform distribution of labels across different clusters in the label space, while simultaneously maximizing the distance from the initial data in the data space. $Q_A$ selects new data such that the labels of the data are uniformly distributed across the label space.

## 3 EXPERIMENT

### 3.1 DATASET AND METRICS

For our experiments, we selected datasets with continuous rather than categorical labels. The first is an uneven synthetic dataset, chosen for intuitive visualization of the results. The second dataset is an airfoil dataset from the UIUC library, simulated using computational fluid dynamics solvers; the labels correspond to the lift-to-drag ratio coefficients, i.e., $\boldsymbol{y} \in \mathbb{R}^1$. The third dataset is a flying wing dataset simulated using computational fluid dynamics solvers; the labels represent the working condition and the lift coefficient, namely $\boldsymbol{y} \in \mathbb{R}^3$ Wang et al. (2025). The fourth dataset is a starship-like dataset Seedhouse (2022), the labels represent the lift coefficient, drag coefficient, pitch moment, and pressure center of the shapes, namely $\boldsymbol{y} \in \mathbb{R}^4$. The geometric models, such as airfoils, flying wings, and starships, are readily available; however, acquiring their corresponding labels necessitates extensive numerical simulations Wu et al. (2024).

Our evaluation framework is designed to measure diversity and accuracy separately, rather than using a combined metric such as FID Yu et al. (2021). Diversity is quantified by a custom variant of the Vendi score Friedman & Dieng (2022), calculated as the average pairwise Euclidean distance of the generated data points. Accuracy is evaluated by the mean squared error of the real labels of generated samples against the given conditions. The labels in our study are derived from distinct sources depending on the dataset: from an analytically designed function in the case of the synthetic dataset, and from numerical simulations for the physical shape datasets, respectively.

$$diversity\ score = \int_{\mathbb{Y}} \mathbb{E}||\boldsymbol{x}_{gen,i} - \boldsymbol{x}_{gen,j}||_2 \, d\boldsymbol{c} \qquad (8)$$

$$accuracy\ score = \int_{\mathbb{Y}} \mathbb{E}(\boldsymbol{c} - \boldsymbol{y}_{gen,i})^2 \, d\boldsymbol{c} \qquad (9)$$

where $\boldsymbol{x}_{gen}$ denotes a generated sample, $\boldsymbol{y}_{gen}$ denotes its corresponding label. Conceptually, both the diversity (Eq8) and accuracy (Eq9) scores are defined directly on the label space $\mathbb{Y}$, within which the Riemann integration is performed for evaluation.

For our experiments, we employed a fully connected neural network with 8 layers and 512 hidden units per layer, using the LeakyReLU activation function. The model was trained with the AdamW optimizer for 4,000,000 steps with a batch size of 512. The learning rate was set to 1e-3 with a decay rate (gamma) of 0.9 applied every 100,000 steps. The model was evaluated over 100 sampling steps.

## 3.2 RESULTS

In each iteration of these tests, 6% of the data is selected. The initial (0-th) round of data selection is performed randomly for all methods, yielding identical start results. For the committee method, SVR, Random Forest, XGBoost, and RBF neural networks are employed to predict the labels of unlabeled data points; the variance of their predictions is then used as the criterion for selecting new samples. The anchor method operates by first selecting a set of fixed anchor conditions and subsequently choosing new data based on the predictive uncertainty estimated under these specific conditions Zhang et al. (2024).

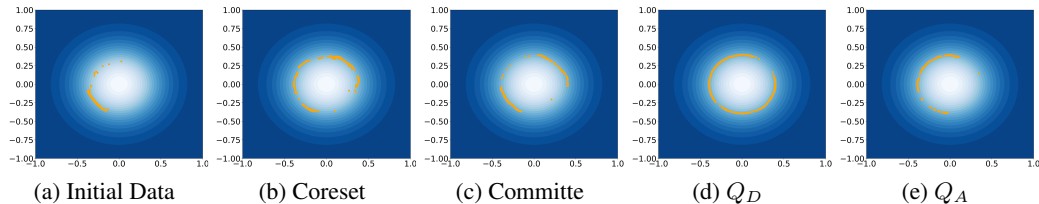

(a) Initial Data  (b) Coreset  (c) Committe  (d) $Q_D$  (e) $Q_A$

Figure 3: Comparison of samples generated given condition 0.5 in the uneven dataset.

Fig3 shows the samples generated by the model under different point selection strategies given condition 0.5. The optimal generation result under condition 0.5 is a circle located at the origin. Fig3a shows model trained on initial data points. It can be seen that due to insufficient data, even on the left half, the generated result is not a complete semicircle. Among all methods, $Q_D$ has the highest diversity, while $Q_A$ has the smallest diversity.

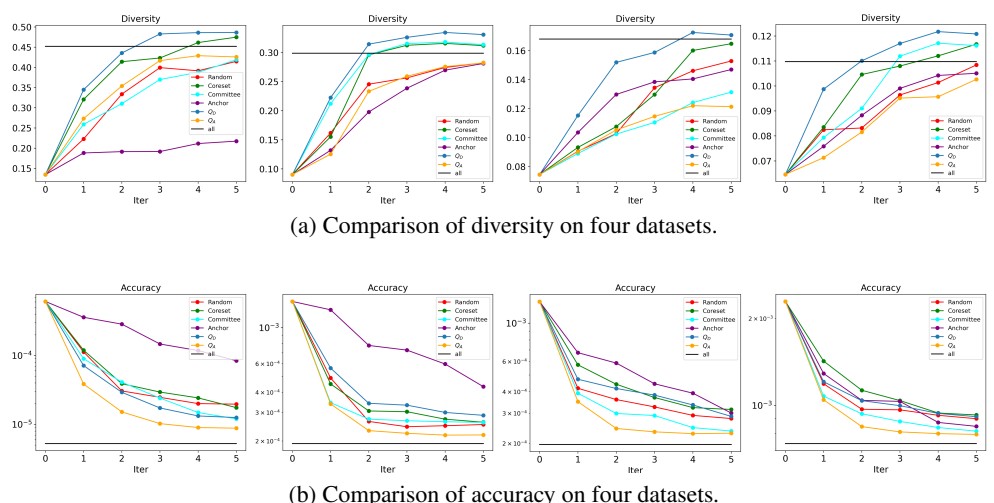

(a) Comparison of diversity on four datasets.

(b) Comparison of accuracy on four datasets.

Figure 4: Comparison of diversity and accuracy on four datasets. The subfigures from left to right correspond to the synthetic, airfoil, flying wing, and starship-like datasets.

Fig4 compares the diversity and accuracy across the four datasets. The results indicate that $Q_D$ achieves the highest diversity, even outperforming the model trained on the full dataset, although this

comes at the cost of reduced accuracy. In contrast, $Q_A$ yields the highest accuracy. The effectiveness of the anchor method is confined to the predefined anchor conditions, and it fails to generalize effectively to conditions outside this set.

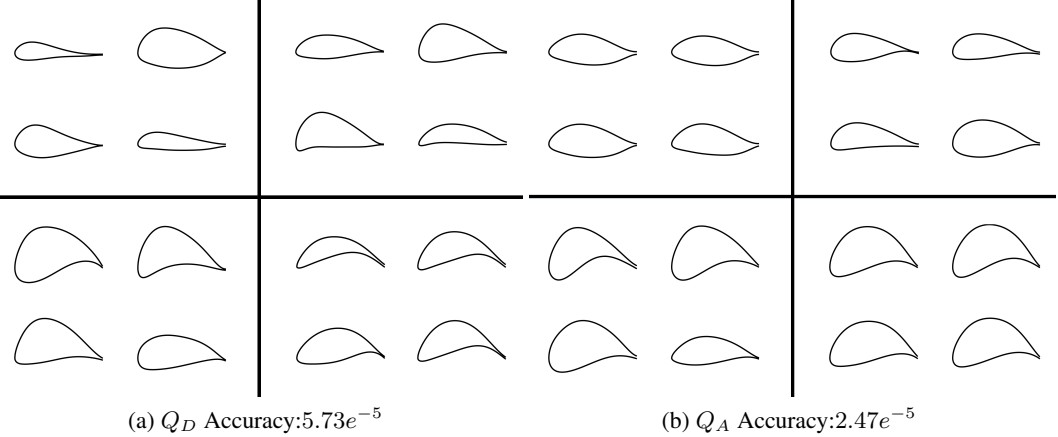

(a) $Q_D$ Accuracy:$5.73e^{-5}$  (b) $Q_A$ Accuracy:$2.47e^{-5}$

Figure 5: Generated airfoil samples under four different conditions. Each panel shows four distinct shapes corresponding to a single condition.

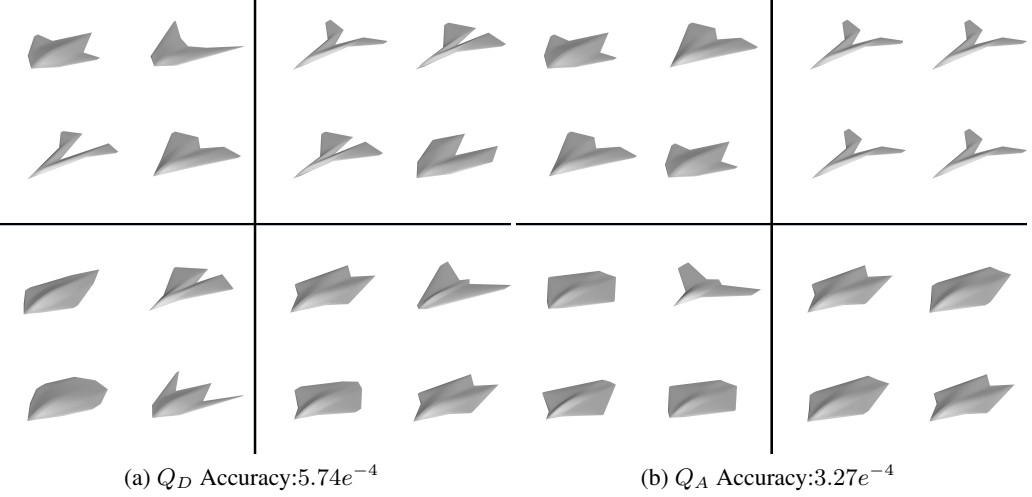

(a) $Q_D$ Accuracy:$5.74e^{-4}$  (b) $Q_A$ Accuracy:$3.27e^{-4}$

Figure 6: Generated flying wing samples under four different conditions. Each panel shows four distinct shapes corresponding to a single condition.

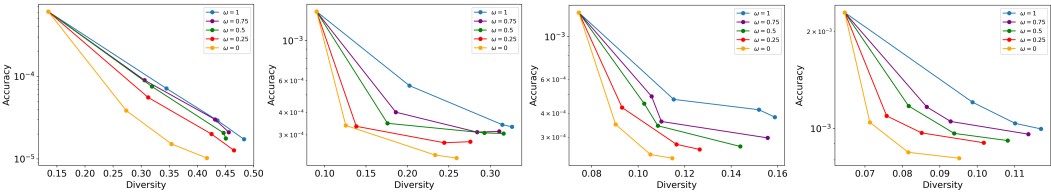

Figure 7: Comparison of different $\omega$ on different datasets.

Fig7 illustrates how the weight $\omega$ in Eq7 can be tuned to control the trade-off between diversity and accuracy: a larger $\omega$ prioritizes diversity, while a smaller $\omega$ favors accuracy. Fig5, Fig6, and Fig8 present a comparison of samples generated by the the model trained under $Q_D$ and $Q_A$ query

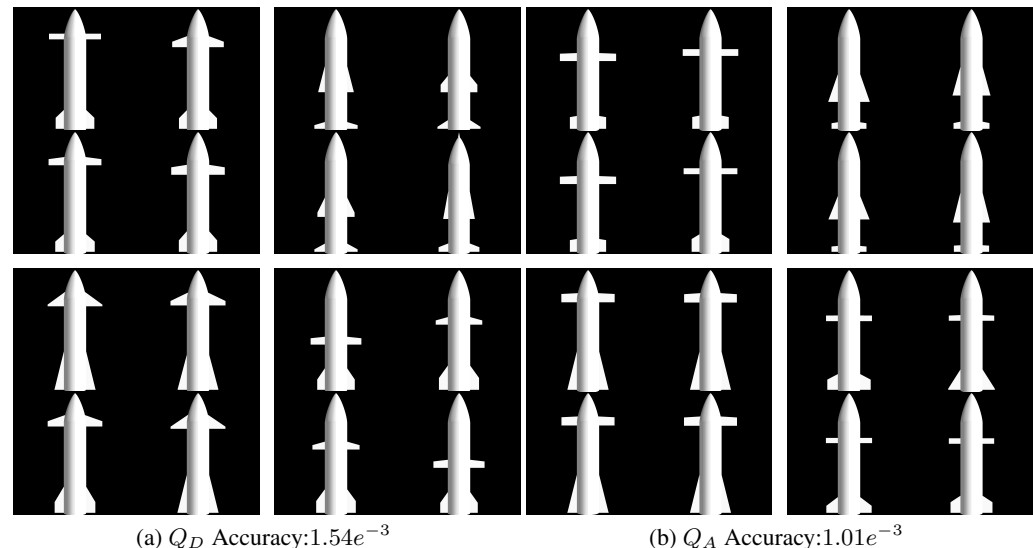

(a) $Q_D$ Accuracy:$1.54e^{-3}$         (b) $Q_A$ Accuracy:$1.01e^{-3}$

Figure 8: Generated starship samples under four different conditions. Each panel shows four distinct shapes corresponding to a single condition.

strategies across different datasets. The results demonstrate that $Q_D$ achieves higher diversity at the cost of lower accuracy, whereas $Q_A$ prioritizes accuracy, resulting in lower diversity.

### 3.3 ABLATION STUDY

The formulation of $Q_D$ comprises three terms, the assessment of the relative impact of each term in Fig9 shows that all three positively influence diversity. The $distance(\mathbf{x}, \mathcal{X})$ term is identified as the most important factor, whereas the $\Delta entropy$ term has a comparatively minor effect.

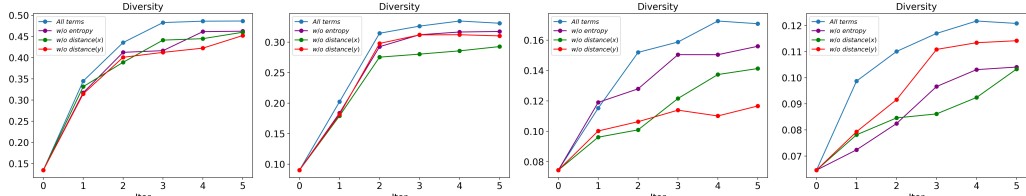

Figure 9: Ablation study on model diversity.

### 4 CONCLUSION AND DISCUSSION

This work tackles active learning for flow matching by first establishing a theoretical foundation via piecewise-linear network and closed-form flow matching models analysis. This framework precisely elucidates the distinct roles of data: label-consistent points drive diversity, while label-varied points bolster accuracy. Capitalizing on this insight, we devise specialized query strategies, one for diversity, the other for accuracy, and a hybrid strategy with adjustable weights to balance them. Comprehensive experiments confirm that our approach surpasses active learning strategies developed for discriminative models. A fundamental characteristic of our approach is its decoupling of the query process from the trained model, relying instead on dataset-level computations. While this allows for efficient allocation of the annotation budget by bypassing the batch-wise process, it also eliminates the need for cumbersome intermediate training cycles. The framework shifts the focus from model-internal diagnostics to data-centric selection, which consequently makes it challenging to directly address or refine the behavioral biases of the final trained model.

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

## A  MATHEMATICAL PROOF

We use the mathematical notation of the flow matching model, $\boldsymbol{x}_0 \in N(\boldsymbol{0}, \boldsymbol{I})$, $\boldsymbol{x}_1$ is data in dataset. $\boldsymbol{u}_t$ is the flow field at time $t$.

**Lemma 1.** *In closed-form flow matching model, if $\boldsymbol{u}_t(\boldsymbol{x}', a_0\boldsymbol{c}_0 + a_1\boldsymbol{c}_1 + ... + a_d\boldsymbol{c}_d) = a_0\boldsymbol{u}_t(\boldsymbol{x}', \boldsymbol{c}_0) + a_1\boldsymbol{u}_t(\boldsymbol{x}', \boldsymbol{c}_1) + ... + a_d\boldsymbol{u}_t(\boldsymbol{x}', \boldsymbol{c}_d)$, then the flow matching model will generate the data in $\{\boldsymbol{x}^* | \boldsymbol{x}^* = a_0\boldsymbol{x}_i + a_1\boldsymbol{x}_j + ... + a_d\boldsymbol{x}_k\}$ when given $\boldsymbol{c}^* = a_0\boldsymbol{c}_0 + a_1\boldsymbol{c}_1 + ... + a_d\boldsymbol{c}_d$. $\boldsymbol{x}_i$ is the data generated by the model when given $\boldsymbol{c}_0$, $\boldsymbol{x}_j$ is the data generated by the model when given $\boldsymbol{c}_1$, and $\boldsymbol{x}_k$ is the data generated by the model when given $\boldsymbol{c}_d$, etc. The dataset contains data labeled as $\boldsymbol{c}_0$, $\boldsymbol{c}_1$, $\boldsymbol{c}_d$, etc.*

*Proof.* In closed-form flow matching model, the generated data is entirely from the dataset. Consider the optimal transmission path $\boldsymbol{x}_t = (1-t)\boldsymbol{x}_0 + t\boldsymbol{x}_1$, and the loss function $\sum ||\boldsymbol{u}_t(\boldsymbol{x}_t, \boldsymbol{c}) - [(1-t)\boldsymbol{x}_0 + t\boldsymbol{x}_1)]||^2$. The flow field at condition $\boldsymbol{c}_0$ is:

$$\boldsymbol{u}_t(\boldsymbol{x}', \boldsymbol{c}_0) = \frac{\sum_i^m p_{t,i}\boldsymbol{e}_{t,i}}{\sum_i^m p_{t,i}} \tag{10}$$

$$\boldsymbol{e}_{t,i} = \frac{\boldsymbol{x}' - t\boldsymbol{x}_{1,i}}{1 - t} \tag{11}$$

$$p_{t,i}(\boldsymbol{x}_1, \boldsymbol{x}') = \frac{1}{(2\pi)^{\frac{d}{2}} |\boldsymbol{\Sigma}_t|^{\frac{1}{2}}} \exp[-\frac{1}{2(1-t)} ||\boldsymbol{e}_{t,i}||^2] \tag{12}$$

$\boldsymbol{e}_{t,i}$ is the noise that make $\boldsymbol{x}_i$ to $\boldsymbol{x}'$ at time $t$. $p_{t,i}(\boldsymbol{x}_1, \boldsymbol{x}')$ is probability density of $\boldsymbol{e}_{t,i}$. $\boldsymbol{x}_i$ is data with label $\boldsymbol{c}_0$ in the dataset, $\boldsymbol{x}_j$ is data with label $\boldsymbol{c}_1$ in the dataset, etc. $m$ is the number of data with label $\boldsymbol{c}_0$, etc, and $n$ is the number of data with label $\boldsymbol{c}_1$ in the dataset, etc.

Thus,

$$\boldsymbol{u}_t(\boldsymbol{x}', a_0\boldsymbol{c}_0 + a_1\boldsymbol{c}_1 + ... + a_d\boldsymbol{c}_d) \tag{13}$$

$$= a_0\boldsymbol{u}_t(\boldsymbol{x}', \boldsymbol{c}_0) + a_1\boldsymbol{u}_t(\boldsymbol{x}', \boldsymbol{c}_1) + ... + a_d\boldsymbol{u}_t(\boldsymbol{x}', \boldsymbol{c}_d) \tag{14}$$

$$= a_0\frac{\sum_i^m p_{t,i}\boldsymbol{e}_{t,i}}{\sum_i^m p_{t,i}} + a_1\frac{\sum_i^n p_{t,j}\boldsymbol{e}_{t,j}}{\sum_j^n p_{t,j}} + ... + a_d\frac{\sum_k^o p_{t,k}\boldsymbol{e}_{t,k}}{\sum_k^o p_{t,k}} \tag{15}$$

$$= \frac{\sum_i^m \sum_j^n ... \sum_k^o p_{t,i}p_{t,j}...p_{t,k}(a_0\boldsymbol{e}_{t,i} + a_1\boldsymbol{e}_{t,j} + ... + a_d\boldsymbol{e}_{t,k})}{\sum_i^m \sum_j^n ... \sum_k^o p_{t,i}p_{t,j}...p_{t,k}} \tag{16}$$

$$= \frac{\sum_i^m \sum_j^n ... \sum_k^o p_{t,i}p_{t,j}...p_{t,k}\frac{(a_0+a_1+...+a_d)\boldsymbol{x}'-t(a_0\boldsymbol{x}_i+a_1\boldsymbol{x}_j+...+a_d\boldsymbol{x}_k)}{1-t}}{\sum_i^m \sum_j^n ... \sum_k^o p_{t,i}p_{t,j}...p_{t,k}} \tag{17}$$

While the vector field directly defined on $\{\boldsymbol{x}^* | \boldsymbol{x}^* = a_0\boldsymbol{x}_i + a_1\boldsymbol{x}_j + ... + a_d\boldsymbol{x}_k\}$ is:

$$\boldsymbol{u}_t^*(\boldsymbol{x}', \boldsymbol{c}^*) = \frac{\sum_l^{nm...o} p_{t,l}\boldsymbol{e}_{t,l}}{\sum_l^{nm...o} p_{t,l}} \tag{18}$$

$$= \frac{\sum_l^{nm...o} p_{t,l}\frac{\boldsymbol{x}'-t(a_0\boldsymbol{x}_i+a_1\boldsymbol{x}_j+...+a_d\boldsymbol{x}_k)}{1-t}}{\sum_l^{nm...o} p_{t,l}} \tag{19}$$

$\boldsymbol{u}_t^*(\boldsymbol{x}', \boldsymbol{c}^*)$ means the model is trained on the interpolation data.

Comparing $\boldsymbol{u}_t(\boldsymbol{x}', a_0\boldsymbol{c}_0 + a_1\boldsymbol{c}_1 + ... + a_d\boldsymbol{c}_d)$ and $\boldsymbol{u}^* +_t (\boldsymbol{x}', \boldsymbol{c}^*)$, we can see that although the two vector fields are not exactly the same, their final generated results are consistent. The difference lies in the different noise schedules they choose.

$\square$

**Lemma 2.** *The sample error for the piecewise-linear neural network driven flow matching model is:*

$$|f(\boldsymbol{x}^*) - \boldsymbol{c}^*| \leq Kmax||\boldsymbol{c}_i - \boldsymbol{c}_j||^2 \tag{20}$$

*Proof.* Consider a subregion of the label space, Its vertices are $\boldsymbol{c}_0$, $\boldsymbol{c}_1$,..., $\boldsymbol{c}_d$. There exists a unique set of weight coefficients for $\boldsymbol{c}^*$ in $d$-dimensional space.

$$[a_0 \quad \cdots \quad a_d] \begin{bmatrix} c_{0,1} & \cdots & c_{0,d} \\ \vdots & \ddots & \vdots \\ c_{d,1} & \cdots & c_{d,d} \end{bmatrix} = [c_1^* \quad \cdots \quad c_d^*]$$

and we get:

$$[a_0 \quad \cdots \quad a_d] = [c_1^* \quad \cdots \quad c_d^*] \begin{bmatrix} c_{0,1} & \cdots & c_{0,d} \\ \vdots & \ddots & \vdots \\ c_{d,1} & \cdots & c_{d,d} \end{bmatrix}^{-1}$$

The sample generated by the model under condition $c^*$ is $x^*$:

$$\begin{bmatrix} x_0^* & \cdots & x_q^* \end{bmatrix} = \begin{bmatrix} a_0 & \cdots & a_d \end{bmatrix} \begin{bmatrix} x_{i,1} & \cdots & x_{i,q} \\ \vdots & \ddots & \vdots \\ x_{k,1} & \cdots & x_{k,q} \end{bmatrix}$$

By using the error bound of linear interpolation, the sample error is:

$$|f(\boldsymbol{x}^*) - \boldsymbol{c}^*| \leq L_f |\boldsymbol{x}^* - \boldsymbol{x}^{gt}| \tag{21}$$

$$= L_f \left| -\frac{1}{2} \sum_{i=0}^{d} a_i (\boldsymbol{c}_i - \boldsymbol{c})^T \boldsymbol{H}_{f^{-1}}(\boldsymbol{\xi}_i)(\boldsymbol{c}_i - \boldsymbol{c}) \right| \tag{22}$$

$$\leq L_f \frac{(d+1)}{2} M max ||\boldsymbol{c}_i - \boldsymbol{c}_j||^2 \tag{23}$$

$$= K max ||\boldsymbol{c}_i - \boldsymbol{c}_j||^2 \tag{24}$$

where $L_f$ is Lipschitz constant. $f(\boldsymbol{x}) = \boldsymbol{y}$ is the real data distribution. $\boldsymbol{x}^{gt}$ is the data with label $\boldsymbol{c}^*$. $f^{-1}$ is the inverse function of $f(x)$ defined on convex hull composed of $\boldsymbol{c}_0, \boldsymbol{c}_1,..., \boldsymbol{c}_d$. $\boldsymbol{H}_{f^{-1}}$ is the Hessian matrix of $f^{-1}$, and $||\boldsymbol{H}_{f^{-1}}|| \leq M$ in the convex hull. $max||\boldsymbol{c}_i - \boldsymbol{c}_j||^2$ represents the maximum distance of any two points in the convex hull.

$\square$

# B   THE USE OF LARGE LANGUAGE MODELS

In this paper, we first manually wrote the paper, then polished it using a large language model, and finally manually calibrated it to avoid the polished results is differ from their original meaning.

