# OpenReview forum: "Active Learning for Flow Matching Model in Shape Design: A Perspective from Continuous Condition Dataset"
_ICLR.cc/2026/Conference — Submitted to ICLR 2026_

### Official Review · Reviewer_cJpe · 2025-10-26

**Soundness:** 2
**Presentation:** 2
**Contribution:** 2
**Rating:** 2
**Confidence:** 3

**Summary:**

The paper investigates the effect of active learning strategies on flow matching. It presents an analysis based on piece-wise linear framework. Based on this analysis, they present an acquisition strategy to increase the diversity of the generative model. Secondly, the authors describe a query strategy to increase the accuracy of the flow matching strategy. The experiments were performed on a number of shape generation datasets. They show that both strategies reach their respective goal of maximizing the diversity or accuracy.

**Strengths:**

1. One of the first works to investigate active learning for flow matching.
2. The writing is easy to follow.

**Weaknesses:**

1. The piece-wise linear framework appears to be overly simplistic. According to the framework, if we would only select a single context c, flow matching could only replicate training data samples?
2. For the diversity strategy $Q_D$ , the terms  $-distance(y, \mathcal{Y})$ and  $\Delta entropy$ seem to be conflicting. The first one is supposed to bring the samples to the label of an already known sample, the second one is supposed to "promote a more uniform label distribution".
3. It appears that the approach of a pool-based setting already reveals all the shapes, as they are the inputs of this unlabeled datasets. Wouldn't we get the perfect diversity by just training the generative model on all the pool inputs, at least unconditionally?
4. The active learning strategies do not make use of the actual flow matching model at all. Instead, the accuracy query strategy is just the standard output diversity maximization, which is not flow-matching specific at all.
6. Many formatting errors (parentheses, spaces around citations, the subfigure captions in Figure 1, referencing equations as eq2 instead of Eq. 2 for example).
7. No statistical evaluation of the results. This is really important for AL, since it can be sensitive to the initial data for example and the small datasets lead to a large variance. Hence the experiments should be repeated multiple times.

**Questions:**

1. How was the coreset baseline applied? Using the latent space of the regression model?
2. In the ablations, did you also try diversity sampling without entropy and distance in label space?
4. Why did you choose an RBF network as the regression model?

---

### Official Review · Reviewer_Hn6d · 2025-10-30

**Soundness:** 2
**Presentation:** 2
**Contribution:** 2
**Rating:** 2
**Confidence:** 3

**Summary:**

This paper describes the application of active learning methods to training flow matching models in a data-efficient way. New selection strategies to adapt standard active learning to this new scenario are proposed, and evaluated. On strategies bases election on diversity, another on accuracy, and a third is a hybrid of the two. This selection strategies are "model-free", relying on characteristics of the data alone rather than the outputs of the model for selection. The proposed method is evaluated using four datasets.

**Strengths:**

The main strengths of the paper are:

-- The work described addresses a niche problem in active learning that is not widely studied and would be of interest to ICLR attendees.
-- The work blends practical and theoretical contributions and insights.
-- The proposed approach is evaluated on multiple datasets.
-- The visual presentation of results is clear and carefully designed.
-- Based on the authors; evaluation experiments the models appear to perform well.
-- The theoretical contributions are supported by detailed appendices.

**Weaknesses:**

The main weaknesses of the paper are:

-- The authors do not sufficiently explain how the active learning problem is mapped to the flow matching, generative model scenario. The exact role of labels in this scenario is not clear from the authors' explanations nor is the role of labelled data in training the models. A much clearer explanation of how the authors approach the active learning problem is required.
-- The paper requires more careful review and revision as there are multiple typographical errors. E.g. “For example, GALISPZhang et al. (2024) consider ”subject of interest” which transforming the open querying problem in the label space into a semi-open one.” and "where et,i is the noise that make xi to x′ “ and "Because d+ 1 points form a d-dimensional plane.” Also, the referencing style appears incorrect and throughout opening and closing quotes in Latex are not used appropriately.
-- The evaluation setup is not completely clear - what data is used for evaluation? Is this different op what is used for training models? Also is the use of accuracy and diversity for evaluation appropriate, given that the proposed approach maximise these?
-- The ideas of model free active learning selection strategies and selection strategies mixing diversity and accuracy exist in the literature. It is not clear exactly where the novelty of the approach lies.

**Questions:**

It would be useful for the authors to answer the following questions:

-- Exactly what is the role of labels in the flow matching model training process described and how does the active learning approach integrate with how that model is trained?
-- Why does accuracy appear to reduce in Figure4 as the algorithm proceed?

---

### Official Review · Reviewer_NJof · 2025-10-31

**Soundness:** 3
**Presentation:** 2
**Contribution:** 2
**Rating:** 2
**Confidence:** 4

**Summary:**

The main contribution is a unique analytical framework to better understand how the composition of a data set affects the behavior of an FM model. Here, the FM neural network is modeled as a Continuous Piecewise-Linear (CPWL) function. From this, the authors derive a crucial finding that data points with labels similar to existing labels provide the main source of diversity in the output of the model whereas data points with dissimilar labels improve the accuracy of the output.

**Strengths:**

It provides a practical process, based on data, which allows to decrease the cost of training superior generative models in critical scientific and engineering fields, which are often label-poor. The essential insight that this selection process can be “decoupled” from the training of the FM model itself, relying solely on properties of the dataset and an inexpensive surrogate model, allows for it to be an efficient and practical process.

**Weaknesses:**

1. The entire theoretical framework is based on the assumption that the FM network behaves as a piecewise linear interpolator. The authors state that their network (8 layer, 512 hidden unit FCN) uses LeakyReLU, which is CPWL. However, this theoretical underpinning rests on what is known as "condensation phenomenon", which is by no means guaranteed to hold for all architectures or training regimes.

2. The diversity strategy $Q_D$ (Eq 4) feels slightly ad-hoc. The first term ($-\text{distance}(y, \mathcal{Y})$) is well-motivated by the theory (Section 2.3). However, the second ($\Delta \text{entropy}$) and third ($\text{distance}(x, \mathcal{X})$) terms are imported from other concepts. The ablation study (Fig 9) then reveals that the entropy term—which is part of the justification for balancing $m$ and $n$ in the 1D case—has a "comparatively minor effect". This makes the final formulation feel more "engineered" than "derived" and slightly undermines the elegance of the data-centric argument.

3. The paper does not discuss the scalability of the proposed methods as the label (condition) dimension $d$ increases. The core analysis in Section 2.3 uses $c \in \mathbb{R}^1$ for intuition, and the experiments go up to $y \in \mathbb{R}^4$. Nonetheless, the theory, e.g. of the error bound (Eq. 5), and entropy better be calculated (clustering necessary) on the partitioning of the label space into convex hulls/simplices, may suffer the curse of dimensionality in the case of intractability of computational solution or irrelevance of the analysis to intractable high-dimensional conditional spaces.

**Questions:**

1. Equation 3’s “generation law” implies that samples generated under new conditions, $c^*$, are basically linear blends of existing training examples. This makes the model sound more like a memorizer and interpolator than a true generator, which raises some concerns about its ability to create genuinely new designs beyond what it has already seen. How do the authors reconcile this interpretation with the well-known creative power of generative models? And could the $Q_A$ strategy—by focusing only on reducing interpolation error—actually discourage the model from being creative?

2. The accuracy strategy $Q_A$ (Eq. 6) is motivated by the error bound in Eq. 5 and works like a coreset method in label space—sampling near the “edges” to shrink the $\max ||c_i - c_j||^2$ term. But this approach could overlook regions where the underlying function $f(x)$ is highly non-linear, even if those regions are relatively small. Have the authors thought about alternative versions of $Q_A$—for example, an uncertainty-based strategy that samples from the center of the largest unexplored region (as shown in Fig. 1b), or from areas with high predicted interpolation error, instead of just focusing on the boundaries?

3. Both $Q_D$ and $Q_A$ rely on a surrogate RBF neural network to predict labels across the entire unlabeled pool. This means the effectiveness of the query strategy heavily depends on how accurate that surrogate is. How sensitive are the results in Figure 4 to the quality of this RBF predictor? Also, what’s the actual computational overhead of retraining this surrogate at each active learning step? It would be helpful to know how this cost compares to the “cumbersome intermediate training cycles” of model-dependent strategies—especially as the unlabeled pool $U^n$ becomes large.

---

### Official Review · Reviewer_voFa · 2025-11-01

**Soundness:** 1
**Presentation:** 1
**Contribution:** 2
**Rating:** 2
**Confidence:** 2

**Summary:**

The paper proposes an active learning method for a flow matching model that emphasizes both accuracy and diversity.

**Strengths:**

Flow matching models are relatively new and it's great this paper has decided to explore active learning for these exciting methods.

**Weaknesses:**

I'll admit that the theoretical foundations of the flow matching model are not clear to me. Additionally, I was having trouble following the piece-wise linear interpolation argument, possibly because I am also not familiar with this theory. What is the condition? What are you interpolating exactly? Could you provide a less technical and more conceptual explanation?

Ignoring my inability to follow the primary theoretical motivation, Equation 4 does not make sense to me. Where do y and Y come from? Additionally, I'm unsure if distance(x, X) is the correct formulation for Coreset optimization (perhaps the greedy version of Coreset?). Adding entropy to the distance metrics seems ad hoc to me. For example, if we were to calculate pairwise distances for the Coreset algorithm and then include entropy when calculating the distances, this seems somewhat more sensible to me. Additionally, there are too many tuning parameters, which suggests that this algorithm probably overfits to a particular choice of tuning parameters. To confirm, in (7), are you combining the different queried points to interpolate? The whole methodology is unclear to me.

Evaluation is also unclear to me - why are you evaluating diversity and accuracy separately and giving them equal weight? We should focus primarily on accuracy and consider diversity as a secondary metric to assess robustness. How we determine accuracy, in simple terms, would also be helpful. How would we assess accuracy if we are generating novel images? It somewhat hurts the evaluation that all the datasets are synthetic and, seemingly, simple, which suggests that the method may not generalize to other use cases. However, I don't fully understand the method, so I may be wrong (perhaps the novelty of the approach warrants looking at simplified evaluations first, but I don't quite understand the method and novelty).

**Questions:**

Please see the "Weaknesses" section.

---

### Meta-Review · Area_Chair_DiGB · 2025-12-17

**Summary:**

Clear rejection - All reviewers lean towards rejection and the authors did not respond.

**Reviewer Concerns:**

The reviewers are concerned about the clarity of the theoretical foundations of the flow matching model used, and the linear framework is over-simplified. Concerns on lacking sufficient experiments have also been brought up.

**Reviewer Scores:**

The authors did not respond throughout the rebuttal discussion phase. No change of scores from reviewers as well.

---

### Decision · Program_Chairs · 2026-01-26

Reject